# CONNECTING GRAPH CONVOLUTION & GRAPH PCA

## ABSTRACT

Graph convolution operator of the GCN model is originally motivated from a localized first-order approximation of spectral graph convolutions. This work stands on a different view; establishing a *mathematical connection between graph convolution and graph-regularized PCA* (GPCA). Based on this connection, the GCN architecture, shaped by stacking graph convolution layers, shares a close relationship with stacking GPCA. We empirically demonstrate that the *unsupervised* embeddings by GPCA paired with a 1- or 2-layer MLP achieves similar or even better performance than many sophisticated baselines on semi-supervised node classification tasks across five datasets including Open Graph Benchmark. This suggests that the prowess of graph convolution is driven by graph based regularization. In addition, we extend GPCA to the (semi-)supervised setting and show that it is equivalent to GPCA on a graph extended with "ghost" edges between nodes of the same label. Finally, we capitalize on the discovered relationship to design an effective initialization strategy based on stacking GPCA, enabling GCN to converge faster and achieve robust performance at large number of layers.

## 1 INTRODUCTION

Graph neural networks (GNNs) are neural networks designed for the graph domain. Since the breakthrough of GCN (Kipf & Welling, 2017), which notably improved performance on the semi-supervised node classification problem, many GNN variants have been proposed; including GAT (Veličković et al., 2018), GraphSAGE (Hamilton et al., 2017), DGI (Veličković et al., 2019), GIN (Xu et al., 2019), PPNP and APPNP (Klicpera et al., 2019), to name a few.

Despite the empirical successes of GNNs in both node-level and graph-level tasks, they remain not well understood due to limited systematic and theoretical analysis of GNNs. For example, researchers have found that GNNs, unlike their non-graph counterparts, suffer from performance degradation with increasing depth, their expressive power decaying exponentially in number of layers (Oono & Suzuki, 2020). Such behavior is only partially explained by the oversmoothing phenomenon (Li et al., 2018; Zhao & Akoglu, 2020). Another surprising observation shows that a Simplified Graph Convolution model, named SGC (Wu et al., 2019), can achieve similar performance to various more complex GNNs on a variety of node classification tasks. Moreover, a simple baseline that does not utilize the graph structure altogether performs similar to state-of-the-art GNNs on graph classification tasks (Errica et al., 2020). These observations call attention to studies for a better understanding of GNNs (NT & Maehara, 2019; Morris et al., 2019; Xu et al., 2019; Oono & Suzuki, 2020; Loukas, 2020; Srinivasan & Ribeiro, 2020). (See Sec. 2 for more on understanding GNNs.)

Toward a systematic analysis and better understanding of GNNs, we establish a connection between the graph convolution operator of GCN (and PPNP) and Graph-regularized PCA (GPCA) (Zhang & Zhao, 2012), and show the similarity between GCN and stacking GPCA. This connection provides a deeper understanding of GCN's power and limitation. Empirically, we also find that GPCA performance matches that of many GNN baselines on benchmark semi-supervised node classification tasks. We argue that the simple GPCA should be a strong baseline in future. What is more, the unsupervised stacking GPCA can be viewed as "unsupervised GCN" and provides a straightforward, yet systematic way to initialize GCN training. We summarize our contributions as follows:

• **Connection between Graph Convolution and GPCA:** We establish the connection between the graph convolution operator of GCN (also PPNP) and the closed-form solution of graph-regularized PCA (GPCA) formulation. We demonstrate that a simple graph-regularized PCA paired with 1- or 2-layer MLP can achieve similar or even better results than state-of-the-art GNN baselines over

several benchmark datasets. We further extend GPCA to (semi-)supervised setting which can generate embeddings using information of labels, which yields better performance on 3 out of 5 datasets. The outstanding performance of simple GPCA supports that the prowess of GCN on node classification task comes from graph based regularization. This motivates the study and design of other graph regularization techniques in the future.

• GPCANET**: New Stacking GPCA model:** Capitalizing on the connection between GPCA and graph convolution, we design a new GNN model called GPCANET shaped by (1) stacking multiple GPCA layers and nonlinear transformations, and (2) fine-tuning end-to-end via supervised training. GPCANET is a generalized GCN model with adjustable hyperparameters that control the strength of graph regularization of each layer. We show that with stronger regularization, we can train GPCANET with fewer (1–3) layers and achieve comparable performance to much deeper GCNs.

• **First initialization strategy for GNNs:** Capitalizing on the connection between GCN and GP-CANET, we design a new strategy to initialize GCN training based on stacking GPCA, outperforming the popular Xaiver initialization (Glorot & Bengio, 2010). We show that the GPCANET-initialization is extremely effective for training deeper GCNs, that significantly improves the convergence speed, performance, and robustness. Notably, GPCANET-initialization is general-purpose and also applies to other GNNs. To our knowledge, it is the first initialization method specifically designed for GNNs.

We open-source code at `http://bit.ly/GPCANet`. All datasets are public-domain.

## 2 RELATED WORK

**Understanding GNNs.** Our work concerns learning on a single graph, hence we limit discussion of related work to node-level GNNs. GCN's graph convolution is originally motivated from the approximation of graph filters in graph signal processing (Kipf & Welling, 2017). NT & Maehara (2019) show that graph convolution only performs low-pass filtering on original feature vectors, and also state a connection between graph filtering and Laplacian regularized least squares. Motivated by the oversmoothing phenomenon of graph convolution, Oono & Suzuki (2020) theoretically prove that GCN can only preserve information of node degrees and connected components when the number of layers goes to infinity, under some conditions of GCN weights. Recently several papers revisited the connection of graph convolution to graph-regularized optimization problem (Li et al., 2019; Ma et al., 2020; Pan et al., 2021; Zhao & Akoglu, 2020; Zhu et al., 2021), which is originally discussed in graph signal processing (Shuman et al., 2013). More specifically, both Ma et al. (2020) and Zhu et al. (2021) relate graph-regularization optimization to several GNNs such as GCN (Kipf & Welling, 2017), APPNP (Klicpera et al., 2019), and GAT (Veličković et al., 2018). However, all previous work study these connections while ignoring the learnable parameters, which are essential for high-performance deep learning. Our work differs from these by establishing a stronger and closer connection to graph-regularized PCA that also takes learnable parameters into account.

**Graph-regularized PCA.** PCA and its variants are standard linear dimensionality reduction approaches. Several work extend PCA to graph-structured data, such as Graph-Laplacian PCA (Jiang et al., 2013) and Manifold-regularized Matrix Factorization (Zhang & Zhao, 2012). For other variants, see Shahid et al. (2016).

**Stacking Models and Deep Learning.** The connection between CNN and stacking PCA has been explored in PCANet (Chan et al., 2015), which demonstrated that the (unsupervised) simple stacking PCA works as well as supervised CNN over a large variety of vision tasks. The original PCANet is shallow and does not have nonlinear transformations, while PCANet+ (Low et al., 2017) overcomes these limitations and pushes the architecture much deeper. The idea of layerwise stacking for feature extraction is not new and was empirically observed to exhibit better representation ability in terms of classification. For a comprehensive review, we refer to Bengio et al. (2013).

**Initialization.** Traditionally, neural networks (NNs) were initialized with random weights generated from Gaussian distribution with zero mean and a small standard deviation (Krizhevsky et al., 2012). As training deeper NNs became extremely difficult due to vanishing gradient and activation functions, Glorot & Bengio (2010) provided a specific weight initialization formula, named Xavier initialization, based on variance analysis without considering activation function. Xavier initialization is widely used for any type of NN even today, and it is the main initialization strategy used for GNNs. Later, He et al. (2015) adapted Xavier initialization to ReLU activation by considering a multiplier. Taking another

direction, Saxe et al. (2013) analyzed the dynamics of training deep NNs and proposed random orthonormal initialization. Mishkin & Matas (2015) further improved orthonormal initialization for batch normalization (Ioffe & Szegedy, 2015). Different from these data-independent approaches, others (Krähenbühl et al., 2016; Seuret et al., 2017; Wagner et al., 2013) have employed data-dependent techniques, like PCA, to initialize deep NNs. Although initialization has been widely studied for general NNs, no specific initialization has been proposed for GNNs. In this work, we propose a data-driven initialization technique (based on GPCA), specific to GNNs for the first time.

# 3 GRAPH CONVOLUTION AND GPCA

## 3.1 GRAPH CONVOLUTION

Consider a node-attributed input graph $G = (V, E, X)$ with $|V| = n$ nodes and $|E| = m$ edges, where $X \in \mathbb{R}^{n \times d}$ denotes the node feature matrix with $d$ features. Broadly, graph convolution operation convolves the features (or representations) over the graph structure.

**GCN.** Similar to other neural networks stacked with repeated layers, GCN contains multiple graph convolution layers each of which is followed by a nonlinear activation. Let $H^{(l)}$ be the $l$-th hidden layer representation, then, each GCN layer performs

$$H^{(l+1)} = \sigma(\tilde{A}_{\text{sym}} H^{(l)} W^{(l)}) \tag{1}$$

where $\tilde{A}_{\text{sym}} = \tilde{D}^{-\frac{1}{2}}(A + I)\tilde{D}^{-\frac{1}{2}}$ denotes the $n \times n$ symmetrically normalized adjacency matrix with self-loops, $\tilde{D}$ is the diagonal degree matrix where $\tilde{D}_{ii} = 1 + \sum_{j=1}^{n} A_{ij}$, $W^{(l)}$ depicts the $l$-th layer parameters (to be learned), and $\sigma$ is the nonlinear activation function. Formally, graph convolution is parameterized with $W$ and maps an input $X$ to a new representation $Z$ as

$$Z = \tilde{A}_{\text{sym}} X W . \tag{2}$$

**PPNP.** For PPNP (Klicpera et al., 2019), the features are first transformed by an MLP before convolving over the graph. Formally, the operation is revised as

$$Z = \mu \left( I - (1 - \mu)\tilde{A}_{\text{sym}} \right)^{-1} \text{MLP}_W(X) = \left( I + \alpha \tilde{L} \right)^{-1} \text{MLP}_W(X) \tag{3}$$

where we replace $\mu$ with $\alpha = (1 - \mu)/\mu$, $\tilde{L} := I - \tilde{A}_{\text{sym}}$ denotes the normalized graph Laplacian, and $W$ depicts the learnable MLP parameters. As matrix inverse is expensive, an approximate version called APPNP that employs the power method (Golub & Van Loan, 1989) is often used in practice.

## 3.2 GRAPH-REGULARIZED PCA (GPCA)

As stated by Bengio et al. (2013), "Although depth is an important part of the story, many *other priors* are interesting and can be conveniently captured when the problem is cast as one of learning a representation." GPCA is one such representation learning technique with a *graph-based prior*.

Standard PCA learns $k$-dimensional projections $Z \in \mathbb{R}^{n \times k}$ of feature matrix $X \in \mathbb{R}^{n \times d}$, aiming to minimize the reconstruction error

$$\|X - ZW^T\|_F^2 , \tag{4}$$

subject to $W \in \mathbb{R}^{d \times k}$ being an orthonormal basis. GPCA extends this formalism to graph-structured data by additionally assuming either smoothing bases (Jiang et al., 2013) or smoothing projections (Zhang & Zhao, 2012) over the graph. In this work we consider the latter case where low-dimensional projections are smooth over the input graph $G$, where $\tilde{L} = I - \tilde{A}_{\text{sym}}$ denotes its normalized Laplacian matrix. The objective formulation of GPCA is then given as

$$\min_{Z,W} \quad \|X - ZW^T\|_F^2 + \alpha \operatorname{Tr}(Z^T \tilde{L} Z) \qquad \text{s.t.} \quad W^T W = I \tag{5}$$

where $\alpha$ is a hyperparameter that balances reconstruction error and the variation of the projections over the graph. Note that the first part of Eq. equation 5, along with the constraint, corresponds to the objective of the original PCA, while the second part is a graph regularization term that aims to "smooth" the learned representations $Z$ over the graph structure. As such, GPCA becomes the standard PCA when $\alpha = 0$.

Similar to PCA, the problem (5) is non-convex but has a closed-form solution (Zhang & Zhao, 2012). Surprisingly, as we show, it has a close connection with the graph convolution formulation in Eq. equation 2. In the following, we give the GPCA solution and then detail its connection to graph convolution.

**Theorem 3.1.** *GPCA with formulation shown in (5) has the optimal solution $(Z^*, W^*)$ following*

$$Z^* = (I + \alpha \tilde{L})^{-1} X W^* \text{ , and} \qquad W^* = (\mathbf{w}_1, \mathbf{w}_2, ..., \mathbf{w}_k) \qquad (6)$$

*where $\mathbf{w}_1, ..., \mathbf{w}_k$ are the eigenvectors of $X^T (I + \alpha \tilde{L})^{-1} X$ corresponding to the largest $k$ eigenvalues.*

*Proof.* The proof can be found in Appendix. A.1. $\qquad\qquad\qquad\qquad\qquad\qquad\qquad\square$

### 3.3 CONNECTION BETWEEN GCN AND GPCA

Let $\Phi_\alpha := I + \alpha \tilde{L}$. The normalized Laplacian matrix $\tilde{L}$ has absolute eigenvalues bounded by 1, thus, all its positive powers have bounded operator norm. When $\alpha \leq 1$, $\Phi_\alpha^{-1}$ can be decomposed into Taylor series as $(I + \alpha \tilde{L})^{-1} = I - \alpha \tilde{L} + \ldots + (-\alpha)^t \tilde{L}^t + \ldots$. The first-order truncated form (i.e. approximation) of the series is

$$(I + \alpha \tilde{L})^{-1} \approx I - \alpha \tilde{L} = (1 - \alpha)I + \alpha \tilde{A}_{\text{sym}} \text{ .} \qquad (7)$$

When $\alpha = 1$, the first-order approximation of $Z^*$ in Theorem 3.1 follows

$$Z^* \approx \tilde{A}_{\text{sym}} X W^* \text{ .} \qquad (8)$$

The (approximate) solution to GPCA in Eq. equation 8 matches the form of graph convolution operation in Eq. equation 2, with $W^*$ plugged in as the eigenvectors of the matrix $X^T \Phi_\alpha^{-1} X$. In other words, there exists some parameter $W^*$ with which GCN becomes the first-order approximation of GPCA.

To reiterate, a key contribution of this work is to show that the graph convolution operation in GCN can be viewed as the first-order approximation of GPCA with $\alpha = 1$ with a learnable $W$. Put differently, the first-order approximation of (unsupervised) GPCA with $\alpha = 1$ can be viewed as a graph convolution with a fixed, data-driven $W$. In other words, Notably, for $\alpha < 1$, Eq. equation 7 shows the connection between GPCA and graph convolution equipped with 1-step (scaled) residual connection.

### 3.4 CONNECTION BETWEEN PPNP AND GPCA

Replacing the MLP in Eq. equation 3 with a single linear layer without activation results in $Z = \left(I + \alpha \tilde{L}\right)^{-1} XW$, which has exactly the same formulation as the solution $Z^*$ in Theorem 3.1 Eq. equation 6. The connection states that the graph convolution in PPNP can be viewed as the GPCA solution with a learnable $W$. Interestingly, the empirical performance improvement of PPNP over GCN (see Table 2 in Klicpera et al. (2019)) may be explained through these connections that they have to GPCA; where PPNP relates to the exact solution of GPCA while GCN is related to its (first-order) approximation.

### 3.5 SUPERVISED GPCA

The standard GPCA problem in (5) is unsupervised. Motiviated from LDA (Balakrishnama & Ganapathiraju, 1998) and PLS (Geladi & Kowalski, 1986), in this section we show how to extend it to the supervised setting, by learning embeddings that not only (1) provide good reconstruction and (2) vary smoothly over the graph structure, but also (3) highly correlate with the response variable(s). For simplicity of presentation, let $\mathbf{z} \in \mathbb{R}^d$ be a 1-d embedding and $Y$ denote the response matrix (in the general case of multiple responses). We write the additional, i.e. (3)rd objective above, as[1]

$$\max_{\mathbf{z}} \quad \left[\text{corr}(Y, \mathbf{z})\right]^T \left[\text{corr}(Y, \mathbf{z})\right] \text{var}(\mathbf{z}) \quad \equiv \quad \max_{\mathbf{z}} \quad \mathbf{z}^T Y Y^T \mathbf{z} \qquad (9)$$

The form of equation 9 (See Appendix. A.3) and the variance-maximizing term $\text{var}(\mathbf{z})$ are for mathematical convenience. Despite agnostic to labels, including $\text{var}(\mathbf{z})$ is intuitive since an implicit objective of data projection (embedding) is to ensure that inherent variation in data is captured as much as possible. In general, we would aim to maximize the trace of $Z^T Y Y^T Z$ for multi-dimensional embeddings.

**Interpretation.** For semi-supervised node classification with $c$ classes, let $\boldsymbol{L} \subset V$ denote the set of labeled nodes. For this task, $Y \in \{0, 1\}^{n \times c}$ would encode the node labels where the $v$-th row of $Y$, denoted $Y_v$, depicts the one-hot encoded label for each $v \in \boldsymbol{L}$. For $u \in V \backslash \boldsymbol{L}$ with unknown labels, $Y_u = \mathbf{0}$, set as the $c$-dimensional all-zero vector. Then, $(YY^T)_{ij}$ is simply equal to 1 when

---

[1]For the optimization to be well-posed, constraints on $\mathbf{z}$ are required, omitted for simplicity of presentation.

nodes $i$ and $j$ share the same label, and otherwise 0 (either because they have different labels or labels are unknown). This term simply enforces the representations $Z_i$ and $Z_j$ of two same-labeled nodes to be similar. In a sense, $YY^T$ adds "ghost" edges between the same-label nodes, further guiding the smoothness of their representations over this extended graph structure. We remark that earlier work (Gallagher et al., 2008) has heuristically introduced edges between same-label nodes to enhance a given graph for the node classification task. In this work, we have derived the theoretical underpinning for this strategy.

**Supervised formulation.** We have shown that requiring the embeddings to correlate with the known labels can be interpreted as additional smoothing over "ghost" edges between the same-label nodes in the graph. As such, we extend the GPCA problem in (5) to the (semi-)supervised setting as

$$\min_{Z,W} \quad \|X - ZW^T\|_F^2 + \alpha \operatorname{Tr}(Z^T \tilde{L}_{\text{spr}} Z) \qquad \text{s.t.} \quad W^T W = I \quad ; \tag{10}$$

$$\text{where } \tilde{L}_{\text{spr}} = I - \tilde{A}_{\text{spr}} , \quad \tilde{A}_{\text{spr}} = (1 - \beta)\tilde{A}_{\text{sym}} + \beta D^{-\frac{1}{2}}(YY^T)D^{-\frac{1}{2}} \tag{11}$$

In Eq. equation 11, $\beta$ is an additional hyperparameter for trading-off the graph-based regularization (i.e. smoothing) due to the actual input graph edges versus the ones introduced through $YY^T$ between the nodes of the same label, and $D$ is the diagonal matrix with $D_{ii} = \sum_{j=1}^n (YY^T)_{ij}$.

**Theorem 3.2.** *Supervised GPCA, as shown in (10) has the optimal solution $(Z^*, W^*)$ following*

$$Z^* = (I + \alpha\tilde{L}_{\text{spr}})^{-1}XW^* , \quad \text{and} \qquad W^* = (\mathbf{w}_1, \mathbf{w}_2, ..., \mathbf{w}_k) \tag{12}$$

*where $\mathbf{w}_1, \ldots, \mathbf{w}_k$ are the top eigenvectors of the matrix $X^T(I + \alpha\tilde{L}_{\text{spr}})^{-1}X$, equivalently $X^T\big((1 + \alpha)I - \big[\alpha(1 - \beta)\tilde{A}_{sym} + \alpha\beta D^{-\frac{1}{2}}YY^T D^{-\frac{1}{2}}\big]\big)^{-1}X$, corresponding to the largest $k$ eigenvalues.*

*Proof.* The proof is similar to that of Theorem 3.1. ∎

### 3.6 APPROXIMATION AND COMPLEXITY ANALYSIS

According to formulations in Theorems 3.1 and 3.2, obtaining $Z^* \in \mathbb{R}^{n \times k}$ and $W^* \in \mathbb{R}^{d \times k}$ requires two demanding computations (1) the inverse of $\Phi_\alpha = (I + \alpha\mathbf{L}) \in \mathbb{R}^{n \times n}$, or in the supervised case $\Phi_\alpha = (I + \alpha\tilde{L}_{\text{spr}})$; and (2) top $k$ eigenvectors of the matrix $X^T\Phi_\alpha^{-1}X \in \mathbb{R}^{d \times d}$. Eigen-decomposition takes $O(d^3)$ (Pan & Chen, 1999), which is scalable as $d$ is usually small. Computing matrix inverse, on the other hand, can take $O(n^3)$ and require $O(n^2)$ memory, which would be infeasible for very large graphs.

To reduce computation and memory complexity, we instead approximately compute $F := \phi_\alpha^{-1}X$, which is a common term for both $Z^*$ and $W^*$. We can equivalently write

$$(I + \alpha\mathbf{L})F = X \implies F + \alpha F = \alpha PF + X \implies F = \frac{\alpha}{1 + \alpha}PF + \frac{1}{1 + \alpha}X$$

for $P = \tilde{A}_{\text{sym}}$ in the unsupervised case and $P = (1 - \beta)\tilde{A}_{\text{sym}} + \beta D^{-\frac{1}{2}}(YY^T)D^{-\frac{1}{2}}$ when supervised.

Then, we can iteratively (with total $T$ iterations) use the power method (Golub & Van Loan, 1989) to compute $F$ as

$$F^{(t+1)} \leftarrow \frac{\alpha}{1 + \alpha}PF^{(t)} + \frac{1}{1 + \alpha}X \tag{13}$$

where $t \in \{0, ..., T\}$ depicts the iteration and $F^{(0)} \in \mathbb{R}^{n \times d}$ is initialized as $X$ (or randomly). For the supervised case, $PF^{(t)}$ is computed through a series of (from right to left) matrix-matrix products. This avoids the explicit construction of matrix $YY^T$ in memory. Overall, solving for $F$ takes $O(T(m + n)d)$ where $m$ is the number of edges in the graph. The supervised case has an additional term $O(Td|\mathbf{L}|c)$ with $c$ being the number of classes and $|\mathbf{L}| \leq n$ be the number of labeled nodes, which can also be upper-bounded by $O(T(m + n)d)$ when treating $c$ as constant.

Having solved for $F$, we perform the matrix-matrix product $Z^* = FW^*$ in $O(ndk)$ and then the eigen-decomposition of $X^T F$ in $O(d^3 + nd^2) = O(nd^2)$ (for $n \geq d$). Assuming $O(d) = O(k)$, overall complexity for computing the 1-layer GPCA is given as $O(Tmd + Tnd + nd^2)$, which is *linear in the number of nodes and edges*. Note that empirically we found $5 \leq T \leq 10$ to be sufficient.

---

**Algorithm 1** GPCANET **Forward Pass and Pre-training**

---
1: **Input:** graph $G = (V, E, X)$, GPCA hyper-parameter(s) $\alpha$ (and $\beta$ if supervised, $\beta = 0$ otherwise), #layers $L$, hidden layer sizes $\{d_1, \ldots, d_L\}$, activation function $\sigma(\cdot)$, #approximation steps $T$
2: **Output:** pre-set layer-wise parameters $\{W^{(1)}, \ldots, W^{(L)}\}$
3: Initialize $H^{(0)} := X$
4: **for** $l = 1$ **to** $L$ **do**
5:     Center $H^{(l-1)}$ by subtracting mean of row vectors
6:     $F \leftarrow H^{(l-1)}$
7:     **for** $t = 1$ **to** $T$ **do**
8:         $PF \leftarrow (1-\beta)\tilde{A}_{\text{sym}}F + \beta D^{-\frac{1}{2}}(YY^T)D^{-\frac{1}{2}}F$
9:         $F \leftarrow \frac{\alpha}{1+\alpha}PF + \frac{1}{1+\alpha}H^{(l-1)}$
10:     **end for**
11:     $W^{(l)} \leftarrow$ top $d_l$ eigenvectors of $H^{{(l-1)}^T}F$
12:     $H^{(l)} \leftarrow \sigma(FW^{(l)})$
13: **end for**

---

## 4 GPCANET: A STACKING GPCA MODEL

### 4.1 GPCANET

Thus far, we drew a connection between the geometrically motivated, manifold-based GPCA and the graph convolution operation of deep NN based GCN. Next we leverage this connection to design a new model called GPCANET that takes advantage of the relative strengths of each paradigm; namely, GPCA's ability to capture data variation and structure, and GCN's ability to capture multiple levels of abstraction (i.e. high-level concepts) through stacked layers and non-linearity.

In a nustshell, GPCANET is a stacking of multiple (unsupervised or supervised) GPCA layers and nonlinear transformations, which shares the same architecture as a multi-layer GCN. It consists of two main stages: (1) **Pre-training**, which *initializes* the layer-wise parameters through closed-form GPCA solutions, and (2) **End-to-end-training**, which *refines* these parameters through end-to-end gradient-based minimization of a global supervised loss criterion at the output layer.

We remark that GPCANET is *not* the same as GCN, as each layer uses the formulation in Thm.s 3.1 and 3.2 (with approximation shown in Sec. 3.6). In fact, when $\alpha = 1$ and $\beta = 0$, GPCANET is the GCN model initialized with GPCANET-initialization, which we discuss more in Sec. 4.2. In other words, GPCANET is a *generalized* GCN model with additional hyperparameters, $\alpha$ and $\beta$, controlling the strength of graph regularization based on the existing or "ghost" edges, respectively.

**Forward Pass and Pre-training stage.** During pre-training, weights of the $l$-th layer, denoted as $W^{(l)} \in \mathbb{R}^{d_{l-1} \times d_l}$, are pre-set (i.e. initialized) as the leading $d_l$ eigenvectors of the matrix $H^{{(l-1)}^T}\Phi_\alpha^{-1}H^{(l-1)}$,[2] where $H^{(l-1)}$ is the representation as output by the $(l-1)$-th layer (with $H^{(0)} := X$), and $\Phi_\alpha$ can be the unsupervised $(I + \alpha \mathbf{L})$ or the supervised $(I + \alpha \tilde{L}_{\text{spr}})$. The pre-training stage takes a single forward pass. Algo. 1 shows both forward pass during end-to-end-training and the pre-training procedure, where line 11 in blue is a step used *only* for pre-training.

*Additional treatment for ReLU:* Nonlinear transformations like ReLU improves model capacity, however at pre-training stage, it causes information loss as all negative values are truncated to 0. This hinders the advantage of using the leading $d_l$ eigenvectors to initialize the weights so as to convey maximum variance (i.e. information) to the next layers. To address this issue, we instead use the leading $d_l/2$ eigenvectors $\{\mathbf{w}_i\}_{i=1}^{d_l/2}$ and their negatives $\{-\mathbf{w}_i\}_{i=1}^{d_l/2}$ to initialize $W^{(l)}$. Empirically we observe this always improves performance when using ReLU activation.

**End-to-end training stage.** Pre-training can be seen as an information-preserving initialization, as compared to an uninformative random initialization, after which we refine the layer-wise parameters via gradient-based optimization w.r.t. a supervised loss criterion at the output layer. Specifically for semi-supervised node classification, we perform an end-to-end training w.r.t. the cross-entropy loss on the labeled nodes. All parameters are updated jointly through backpropagation during this stage, with forward computation shown in Algo.1 (excluding line 11).

---

[2]If $d^{(l)}$ is greater than the number of eigenvectors, all eigenvectors are used, with additional vectors generated from random projection of eigenvectors.

## 4.2 GPCANET-INITIALIZATION FOR GCN

When we set $\alpha = 1$, $\beta = 0$, and approximate the matrix inverse $(I + \alpha \mathbf{L})^{-1}$ via first-order truncated Taylor expansion as shown in Eq. equation 7 , GPCANET has the same architecture with GCN. As such, we can use the pre-training stage of GPCANET to initialize GCN with only minor modification. Specifically, we replace lines 6 through 10 in Algo. 1 with the following single line:

$$F \leftarrow \tilde{A}_{\text{sym}} H^{(l-1)} \tag{14}$$

The modified initialization is for GCN and is driven by the mathematical connection between GPCANET and GCN that we established. We expect that adapting it for other GNNs is also possible although we do not pursue this direction here.

## 5 EXPERIMENTS

In this section we design extensive experiments to answer the following questions. (**Q1**) How does the simple, *unsupervised and shallow* GPCA compare to its multi-layer extension GPCANET, as well as to existing GNNs? (**Q2**) How does our extended, semi-supervised GPCA compare to the original, unsupervised GPCA? (**Q3**) Does GPCANET-initialization improve GCN accuracy and robustness?

### 5.1 EXPERIMENTAL SETUP

**Datasets.** We focus on semi-supervised node classification (SSNC) and use 5 benchmark datasets: First three, CORA, CITESEER, PUBMED (Sen et al., 2008), are relatively small ($2K$ to $10K$ nodes) but widely-used citation graphs. For these we use the data splits in Kipf & Welling (2017). The others, ARXIV and PRODUCTS, are newest and much larger ($100K$ to $2000K$) node classification benchmarks from Open Graph Benchmark (Hu et al., 2020), for which we use the official data splits. Data statistics can be found in Appendix. A.4.

**Baselines.** We compare (unsupervised & semi-supervised) GPCA and GPCANET to state-of-the-art (SOTA) GNNs, including **GCN** (Kipf & Welling, 2017), **APPNP** (Klicpera et al., 2019), **GAT** (Veličković et al., 2018), and GraphSAGE (**G-SAGE**) (Hamilton et al., 2017).

**Model configuration and training.** For each dataset, we define a separate pool of values for the hyperparameters (HPs): learning rate, weight decay, number of layers, hidden size, dropout rate, and regularization trade-off terms $\alpha, \beta$. For fair comparison, all models share the *same* HP pools during training. See Appendix. A.5 for HP configurations and other details.

### 5.2 Q1: PERFORMANCE OF (UNSUPERVISED) GPCA AND GPCANET

**GPCA.** Having proved the mathematical connection between GPCA, GCN, and PPNP, we expect unsupervised GPCA ($\beta = 0$) to generate comparable representations. We perform GPCA with different $\alpha \in \{1, 5, 10, 20, 50\}$ (Appendix. Table 5) to obtain node representations and pass those to a 1- or 2-layer MLP. We compare to GCN, APPNP, as well as other GNNs; GAT and G-SAGE.

The performance results are given in Table 1. Due to the scale of the largest two datasets, ARXIV and PRODUCTS, we list the reported performance at OGB-leaderboard[3] (depicted by *) for G-SAGE on both datasets, and that of (Cluster-)GAT on PRODUCTS.

We find that the simple 1-layer GPCA paired with MLP performs consistently better than the multi-layer GCN model across all 5 datasets. GPCA's performance is also comparable to or better than other SOTA GNNs. This is quite notable, given that GPCA is not only shallow but also unsupervised, whereas all other baselines are trained end-to-end, and with the exception of APPNP, they exhibit a multi-layer architecture. By carefully looking at the performance of GPCA with varying $\alpha$ (see Appendix. A.6), we find that different datasets have different best selected $\alpha^*$ (in Table 1 top to bottom: $\alpha^* = \{50, 5, 10, 20, 20\}$) but in general a relatively larger $\alpha$ (compared to graph convolution of GCN that is equivalent to $\alpha = 1$) is preferable for all datasets. Larger $\alpha$ implies stronger graph-regularization on the representations. The outstanding performance of the simple GPCA empirically confirms that the power of GNNs on the SSNC problem is mainly driven by graph regularization.

**GPCANET.** Compared to the 1-layer GPCA, GPCANET has a deeper architecture along with nonlinear activation function. Moreover, it employs hyperparameter $\alpha$ at every layer to control the degree of graph regularization. As each graph convolution has fixed level of graph regularization, one may hypothesize that increasing the number of layers ($L$) corresponds to increasing the degree

---

[3]https://ogb.stanford.edu/docs/leader_nodeprop/

Table 1: Comparison btwn. unsupervised GPCA ($\beta = 0$), GPCANET, and existing (supervised) SOTA GNNs on 5 datasets, w.r.t. mean test accuracy and standard deviation (in parentheses) over 5 different seeds. Those marked with $*$ are reported values at the OGB-leaderboard[3]. Highest mean performance is **in bold** and the second highest is underlined.

| | GPCA | GPCANET | GCN | APPNP | GAT | G-SAGE |
|---|---|---|---|---|---|---|
| CORA | 81.10 (0.00) | 80.64 (0.33) | 80.62 (0.90) | 81.35 (0.18) | 79.27 (0.50) | **81.48** (0.83) |
| CITESEER | **71.80** (0.75) | 71.36 (0.21) | 71.25 (0.05) | 70.33 (0.75) | 69.65 (0.59) | 71.20 (0.92) |
| PUBMED | 78.78 (0.36) | 78.52 (0.17) | 78.42 (0.25) | **78.95** (0.36) | 78.23 (0.54) | 77.78 (0.29) |
| ARXIV | 71.86 (0.18) | **72.20** (0.15) | 70.64 (0.17) | 70.55 (0.27) | 71.11 (0.11) | 71.49$^*$(0.27) |
| PRODUCTS | 79.23 (0.14) | **80.05** (0.29) | 77.90 (0.33) | 77.96 (0.34) | 79.23$^*$(0.78) | 78.29$^*$(0.16) |

of graph regularization. We empirically test this hypothesis using GPCANET, by varying both $L$ (2 to 10) and $\alpha$ (0.1 to 10) to show their connection (hidden size is fixed as 128). The result is shown in Figure 1. The diagonal pattern (in dark blue) empirically suggests that increasing the number of layers has the same effect as increasing graph regularization via $\alpha$.

The corresponding interaction between $\alpha$ and number of layers suggests that we can train a GPCANET with fewer number of layers yet achieve similar regularization by increasing $\alpha$. Such a shallow model that in fact behaves like a deep one has the advantage of less memory requirement and faster training due to fewer parameters.

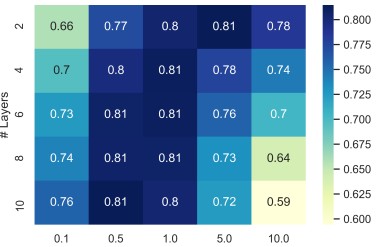

To this end, we train 1–3-layer GPCANET with varying $\alpha$ (higher $\alpha$'s for fewer layers, see Appendix. A.7), and select the best $\alpha$ and number of layers using validation set. We report test set performance in Table 1. We do not observe much improvement by GPCANET over other models on smaller datasets CORA, CITESEER, PUBMED, but notable gains on the larger ARXIV and PRODUCTS. As such, GPCANET enables shallow model training via tunable hyperparameter $\alpha$, achieving comparable or better performance.

Figure 1: GPCANET performance (avg. over 5 seeds) with varying number of layers ($L$) and $\alpha$ on CORA. Increasing $L$ has similar effect as increasing $\alpha$. Results also hold for the other datasets.

### 5.3 Q2: UNSUPERVISED VS. SEMI-SUPERVISED GPCA

The representations generated by unsupervised GPCA does not use any label information from training data. In this work, we have extended GPCA to (semi-)supervised setting with an additional HP, namely $\beta \in [0, 1]$ that trades-off graph regularization due to the actual input graph edges versus the "ghost" ones added through $YY^T$. Overfitting can hurt performance when $\beta$ is too large or when there is a distribution shift between the training and test sets. For ARXIV and PRODUCTS, we empirically observe that $\beta > 0$ always degrades performance, possibly because of the distribution difference between the training and test sets as described in OGB (Hu et al., 2020). Therefore we only study the effect of $\beta$ on CORA, CITESEER and PUBMED. The pool for $\beta > 0$ is $\{0.1, 0.2\}$.

Table 2: Comparison btwn. Supervised (S-)GPCA ($\beta > 0$) and Unsupervised (U-)GPCA ($\beta = 0$), w.r.t. mean test accuracy and standard deviation (in parentheses) over 5 different seeds. Also shown (bottom row) is the performance by the best method in Table 1. Highest mean performance is highlighted **in bold**.

| | CORA | CITESEER | PUBMED |
|---|---|---|---|
| U-GPCA | 81.10 (0.00) | 71.80 (0.75) | 78.78 (0.36) |
| S-GPCA (ALL $\beta > 0$) | 81.17 (0.27) | **73.20** (0.71) | **79.40** (0.69) |
| S-GPCA $\beta = 0.1$ | 81.17 (0.27) | 72.07 (0.37) | **79.40** (0.69) |
| S-GPCA $\beta = 0.2$ | **81.90** (0.00) | **73.20** (0.71) | 78.73 (0.59) |
| TABLE 1 BEST | 81.48 (0.83) | 71.80 (0.75) | 78.95 (0.36) |

Results are shown in Table 2, where (ALL $\beta > 0$) depicts the selected configuration for which S-GPCA achieves highest validation accuracy. The performance of the best method in Table 1, respectively of G-SAGE, (unsupervised) GPCA, and APPNP, is also shown for comparison. Notably, supervised GPCA provides a slight gain over unsupervised GPCA across all 3 datasets, which also improves over the competing baseline methods.

## 5.4 Q3: GPCANET-INITIALIZATION FOR GCN

Finally, we evaluate the effectiveness of GPCANET-initialization for GCN in terms of performance and robustness under different model sizes, i.e. number of layers $L$ or number of training parameters. For comparison, Xavier initialization (Glorot & Bengio, 2010) is also used to initialize GCN.

Table 3: Test set performance of GCN with Xaiver- versus GPCANET-initialization, w.r.t. varying number of layers ($L$) across all datasets. Each reported value is based on the best selected configuration on validation data. GPCANET-init. enables higher performance that is also stable with increasing depth.

| DATASET | $L$=2 | $L$=3 | $L$=5 | $L$=10 | $L$=15 |
|---|---|---|---|---|---|
| CORA XAIVER-INIT | 80.62 | **80.62** | 79.40 | 76.37 | 66.07 |
| CORA GPCANET-INIT | **81.67** | 79.50 | **80.90** | **79.82** | **78.00** |
| CITESEER XAIVER-INIT | 71.25 | **70.15** | **71.10** | 61.90 | 57.40 |
| CITESEER GPCANET-INIT | **71.27** | 69.27 | 70.15 | **68.67** | **67.87** |
| PUBMED XAIVER-INIT | **78.42** | **77.90** | 77.07 | 77.00 | 45.80 |
| PUBMED GPCANET-INIT | 78.05 | 77.25 | **78.07** | **77.80** | **78.03** |
| ARXIV XAIVER-INIT | 69.61 | **70.64** | 70.33 | 68.32 | 61.68 |
| ARXIV GPCANET-INIT | **69.76** | 70.72 | **70.52** | **69.77** | **66.28** |
| PRODUCTS XAIVER-INIT | 77.90 | 78.65 | 78.08 | 76.27 | 74.70 |
| PRODUCTS GPCANET-INIT | **78.13** | **78.71** | **78.22** | **77.47** | **75.90** |

We report the test set performance (averaged over 5 seeds) of the GCN model using both initializations in Table 3. The results show that GPCANET-initialization tends to outperform the widely-used Xavier initialization. The improvement grows with increasing number of layers, which is significant at large depths. Notably, GCN with GPCANET-initialization exhibits stable performance across all layers.

Besides looking at the average performance, we further study whether GPCANET-initialization improves the training robustness, by reducing performance variation across different seeds. To this end, we first choose the best configuration for each initialization method based on validation performance, and train the GCN model with the chosen configuration using 100 random seeds.

In Figure 2 we present the histogram of test set accuracy over 100 runs with different seeds for ARXIV. (For results on other datasets, see Appendix. A.8.) For both 2-layer and 15-layer GCN, GPCANET-initialization not only outperforms Xavier-initialization w.r.t. average performance, but also in terms of robustness, achieving much lower performance

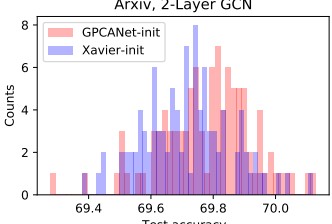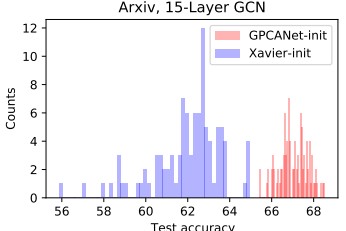

Figure 2: Comparison between Xavier-init. and GPCANET-init. in terms of test accuracy robustness over 100 seeds on ARXIV. GPCANET-init. enables robust training especially at larger depth.

variation and few bad outliers, especially for deeper GCN. As such, it acts as a strong data-driven prior, facilitating the training of numerous parameters across many layers by identifying a promising region of the parameter space from which supervised fine-tuning is initiated.

## 6 CONCLUSION

In this work we have (1) discovered a mathematical connection between GPCA and graph convolution of GCN and PPNP; (2) extended GPCA to the (semi-)supervised setting; (3) proposed GPCANET, by stacking GPCA and nonlinear activation, which is a generalized GCN model with an additional hyperparameter to control the degree of graph regularization, and (4) introduced the GPCANET-initialization based on the established connection. Accordingly, we designed extensive experiments demonstrating that ($i$) the unsupervised shallow GPCA achieves comparable or better performance than GCN, APPNP, as well as other modern GNNs which suggests that graph convolution's power is mainly driven by graph regularization; ($ii$) semi-supervised GPCA helps improve performance and should be a powerful yet simple baseline in future research; ($iii$) GPCANET enables the training of shallow models with competitive performance via increasing the degree of graph regularization at each layer, with reduced memory and training time cost; and finally ($iv$) GPCANET-initialization acts as a strong data-driven prior for GCN training, enabling robust performance. Our methodological contributions ( 3) & 4) above) capitalize on the discovery of our theoretical findings ( 1) & 2) ), shedding new light toward a better understanding and design of GNNs.

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

# A   APPENDIX

## A.1   PROOF OF THEOREM 3.1

*Proof.* We give the proof in two steps.

*Step 1: For a fixed $W$, Solve optimal $Z^*$ as a function of $W$:* When fixing $W$ as constant, the problem becomes quadratic and convex. There is a unique solution, given by first-order optimal condition. Let $\ell$ denote the objective function as given in equation 5. Its gradient can be calculated as

$$\frac{\partial \ell}{\partial Z} = 2(I + \alpha \tilde{L})Z - 2XW \ . \tag{15}$$

Setting equation 15 to 0 leads to the solution $Z^* = (I + \alpha \tilde{L})^{-1}XW$.

*Step 2: Replace $Z$ with $Z^*$, Solve optimal $W^*$:* Substituting $Z$ in objective $\ell$ with $Z^* = (I + \alpha \tilde{L})^{-1}XW$, we reduce the optimization to

$$\min_{W, W^T W = I} \ \|X - (I + \alpha \tilde{L})^{-1}XWW^T\|_F^2 \ + \ \alpha \operatorname{Tr}\left[W^T X^T (I + \alpha \tilde{L})^{-1}\tilde{L}(I + \alpha \tilde{L})^{-1}XW\right] \ . \tag{16}$$

For this part only, let $M = (I + \alpha \tilde{L})^{-1}$ to simplify notation. We can show that equation 16 is equivalent to

$$\min_{W, W^T W = I} \ \operatorname{Tr}(XX^T + MXWW^T WW^T X^T M)$$

$$- 2\operatorname{Tr}(MXWW^T X^T) + \alpha \operatorname{Tr}(W^T X^T M\tilde{L}MXW) \tag{17}$$

Using the cyclic property of (Tr)ace (and plugging $(I + \alpha \tilde{L})^{-1}$ for $M$ back), we can write it as (see Supp. A.2 for detailed derivation.)

$$\max_{W, W^T W = I} \ \operatorname{Tr}\left[W^T X^T (I + \alpha \tilde{L})^{-1}XW\right] \ . \tag{18}$$

Based on the spectral theorem of PSD matrices, the optimal solution $W^*$ of problem equation 18 is the combination of eigenvectors, associated with the largest $c$ eigenvalues of the graph-revised covariance matrix $X^T (I + \alpha \tilde{L})^{-1}X$. $\qquad \square$

## A.2   DERIVATION FROM EQ. EQUATION 17 TO EQ. EQUATION 18

For this part only, let $A = (I + \alpha \tilde{L})^{-1}$ to simplify the notation. We can show that equation 16 is equivalent to

$$\min_{W, W^T W = I} \ \operatorname{Tr}(XX^T) - 2\operatorname{Tr}(AXWW^T X^T)$$

$$+ \operatorname{Tr}(AXWW^T WW^T X^T A) + \alpha \operatorname{Tr}(W^T X^T A\tilde{L}AXW)$$

$$\equiv \max_{W, W^T W = I} \ 2\operatorname{Tr}(AXWW^T X^T) - \operatorname{Tr}(AXWW^T X^T A)$$

$$- \alpha \operatorname{Tr}(W^T X^T A\tilde{L}AXW) \tag{19}$$

Using the cyclic property of (Tr)ace, we can write

$$\max_{W, W^T W = I} \ 2\operatorname{Tr}(W^T X^T AXW) - \operatorname{Tr}(W^T X^T AAXW)$$

$$- \alpha \operatorname{Tr}(W^T X^T A\tilde{L}AXW)$$

$$\max_{W, W^T W = I} \ \operatorname{Tr}\left[W^T X^T (2A - AA - A(\alpha \tilde{L})A)XW\right]$$

$$\max_{W, W^T W = I} \ \operatorname{Tr}\left[W^T X^T (A + \{I - A(I + \alpha \tilde{L})\}A)XW\right]$$

$$\max_{W, W^T W = I} \ \operatorname{Tr}\left[W^T X^T (I + \alpha \tilde{L})^{-1}XW\right] \tag{20}$$

where the objective simplifies upon replacing $A$ with $(I + \alpha \tilde{L})^{-1}$.

## A.3 Derivation of Equivalence in Eq. equation 9

$$\max_{\mathbf{z}} \quad \big[\mathrm{corr}(Y, \mathbf{z})\big]^T \big[\mathrm{corr}(Y, \mathbf{z})\big] \mathrm{var}(\mathbf{z})$$

$$\equiv \max_{\mathbf{z}} \quad \mathrm{var}(Y)\big[\mathrm{corr}(Y, \mathbf{z})\big]^T \big[\mathrm{corr}(Y, \mathbf{z})\big] \mathrm{var}(\mathbf{z}) \tag{21}$$

$$\equiv \max_{\mathbf{z}} \quad \big[\mathrm{cov}(Y, \mathbf{z})\big]^T \big[\mathrm{cov}(Y, \mathbf{z})\big] \tag{22}$$

$$\text{where } \mathrm{cov}(Y, \mathbf{z}) = \sqrt{\mathrm{var}(Y)}\mathrm{corr}(Y, \mathbf{z})\sqrt{\mathrm{var}(\mathbf{z})}$$

$$\equiv \max_{\mathbf{z}} \quad \big[Y^T\mathbf{z}\big]^T \big[Y^T\mathbf{z}\big] \tag{23}$$

$$\equiv \max_{\mathbf{z}} \quad \mathbf{z}^T Y Y^T \mathbf{z} \tag{24}$$

Note that in equation 21 we added the term $\mathrm{var}(Y)$ without affecting the optimization problem as it is with respect to $\mathbf{z}$.

## A.4 Dataset Statistics

Table 4: Statistics of used datasets.

| Dataset | #Nodes | #Edges | #Features | #Classes | Train/Val./Test |
|---|---|---|---|---|---|
| Cora | 2,708 | 5,429 | 1,433 | 7 | 5.2%/18.5%/36.9% |
| CiteSeer | 3,327 | 4,732 | 3,703 | 6 | 3.6%/15%/30% |
| PubMed | 19,717 | 44,338 | 500 | 3 | 0.3%/2.5%/5% |
| Arxiv | 169,343 | 1,166,243 | 128 | 40 | 54%/18%/28% |
| Products | 2,449,029 | 61,859,140 | 100 | 47 | 8%/2%/90% |

Datasets used in the experiments are presented in Table 4. Cora, CiteSeer, and PubMed can be downloaded in Pytorch Geometric Library Fey & Lenssen (2019). Arxiv and Products can be accessed in `https://ogb.stanford.edu/`.

## A.5 Hyperparameter Configurations

We setup hyperparameters pool for each dataset, presented in Table 5. All methods use the *same* pool. The only exception is GPCA, as GPCA is just a 1-layer shallow model which can be trained with lager learning rate; we use 0.1 learning rate for it on all datasets.

Table 5: Hyperparameters pool for each dataset, includes learning rate (LR), weight decay (WD), number of layers (#Layers), hidden size, dropout, $\alpha$, and $\beta$. For Arxiv and Products, weight decay is set as 0 because the dataset is large and no overfit happened. Same reason for choosing smaller dropout rate for them.

| Dataset | LR | WD | #Layers | Hidden |
|---|---|---|---|---|
| Cora | 0.001 | [0.0005, 0.005, 0.05] | [2, 3, 5, 10, 15] | [128, 256] |
| CiteSeer | 0.001 | [0.0005, 0.005, 0.05] | [2, 3, 5, 10, 15] | [128, 256] |
| PubMed | 0.001 | [0.0005, 0.005, 0.05] | [2, 3, 5, 10, 15] | [128, 256] |
| Arxiv | 0.005 | 0 | [2, 3, 5, 10, 15] | [128, 256] |
| Products | 0.001 | 0 | [2, 3, 5, 10, 15] | [128, 256] |

| Dataset | Dropout | $\alpha$ | $\beta$ |
|---|---|---|---|
| Cora | [0, 0.5] | [1, 5, 10, 20, 50] | [0, 0.1, 0.2] |
| CiteSeer | [0, 0.5] | [1, 5, 10, 20, 50] | [0, 0.1, 0.2] |
| PubMed | [0, 0.5] | [1, 5, 10, 20, 50] | [0, 0.1, 0.2] |
| Arxiv | [0, 0.2] | [1, 5, 10, 20, 50] | 0 |
| Products | [0, 0.1] | [1, 5, 10, 20, 50] | 0 |

Models are trained on every configuration across HP pools and picked based on validation performance. We use the Adam optimizer for all models. Learning rate is first manually tuned for each dataset to achieve stable training, and the same learning rate is fixed for all models—we empirically

observed that learning rate is sensitive to datasets but insensitive to models. For GPCA and GP-CANET, number of power iterations in Eq. equation 13 is always set to 5. All experiments use the maximum training epoch as 1000 and repeat 5 times. Detailed configuration of HPs can be found in Supp. A.5. We mainly use a single GTX-1080ti GPU for small datasets CORA, CITESEER, and PUBMED. RTX-3090 GPU is used for ARXIV and PRODUCTS.

**Mini-batch training.** As nodes are not independent, GNN is mostly trained in full-batch under semi-supervised setting. We use full-batch training for all datasets except PRODUCTS, which is too large to fit into GPU memory during training. ClusterGCN Chiang et al. (2019), a subgraph based mini-batch training algorithm, is used to train GCN and GPCANET. For evaluation, we still use full-batch since a single forward pass can be conducted without memory issues. Initialization is also employed in full-batch.

**Fair evaluation.** Instead of picking the hyperparameter configurations manually, reported (test) performance is based on the *best* configuration selected using validation performance, where all models leverage the *same* hyperparameter pools. Further, each configuration from the pool is conducted 5 times to reduce randomness.

### A.6 GPCA WITH VARYING $\alpha$

Table 6: Performance of unsupervised GPCA ($\beta = 0$) for varying $\alpha$ w.r.t. mean test accuracy and standard deviation (in parentheses). GPCA (best $\alpha$) selects $\alpha \in \{1, 5, 10, 20, 50\}$ based on validation, whereas GPCA with specific $\alpha$ uses the specified fixed $\alpha$.

|  | CORA | CITESEER | PUBMED | ARXIV | PRODUCTS |
|---|---|---|---|---|---|
| GPCA (BEST $\alpha$) | 81.10 (0.00) | 71.80 (0.75) | 78.78 (0.36) | 71.86 (0.18) | 79.23 (0.14) |
| GPCA-$\alpha$=1 | 72.57 (0.79) | 70.90 (0.58) | 76.92 (0.30) | 65.47 (0.26) | 73.65 (0.07) |
| GPCA-$\alpha$=5 | 80.95 (0.17) | 71.80 (0.75) | **79.40** (0.29) | 70.69 (0.11) | 78.66 (0.09) |
| GPCA-$\alpha$=10 | **82.23** (0.58) | 71.65 (0.53) | 78.78 (0.36) | 71.37 (0.09) | **79.24** (0.09) |
| GPCA-$\alpha$=20 | 82.05 (0.54) | **72.15** (0.47) | 78.15 (0.50) | **71.86** (0.18) | 79.23 (0.14) |
| GPCA-$\alpha$=50 | 81.10 (0.00) | 71.50 (0.32) | 78.00 (0.19) | 71.48 (0.15 | 78.92 (0.10) |

### A.7 CONFIGURATIONS FOR EXPERIMENTS OF 1~3-LAYER GPCANET

To train a shallow GPCANET with tunable $\alpha$ ($\beta$=0 is used), we setup different $\alpha$ pool for different number of layers, because the effect of increasing $\alpha$ is the same to increasing number of layers (shown in Figure 1). We report the pool for $\alpha$ for each layer in Table 7. For other parameters we use the same setting mentioned in Table 5.

Table 7: Pool of $\alpha$ for 1~3-layer GPCANET, same across all datasets.

| # LAYERS | POOL OF $\alpha$ |
|---|---|
| 1-LAYER | [10, 20, 30] |
| 2-LAYER | [3, 5, 10] |
| 3-LAYER | [1, 2, 3, 5] |

### A.8 GPCANET-INIT'S ROBUSTNESS FOR ADDITIONAL DATASETS

Histogram of test set accuracy over 100 runs for GCN initialized by Xavier-initialization and GPCANET-initialization in CORA (Figure 3), CITESEER (Figure 4), and PUBMED (Figure 5). We have ignored PRODUCTS as it takes too long to run 100 times, but the result should be similar.

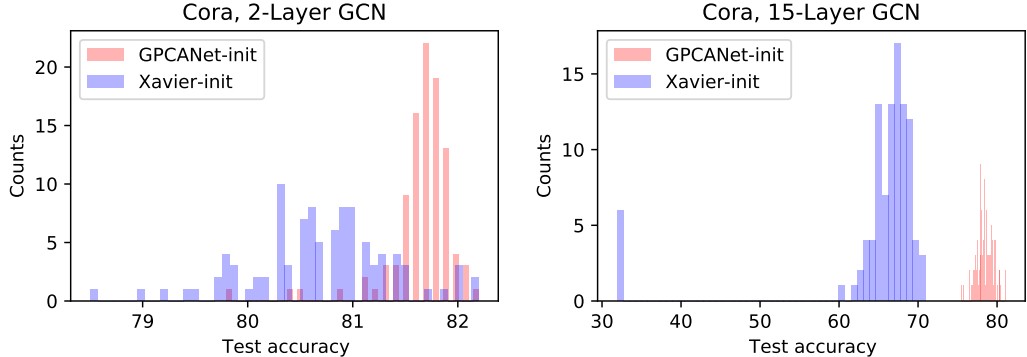

Figure 3: Comparison between Xavier-init and GPCANET-init in terms of test accuracy robustness over 100 seeds on CORA.

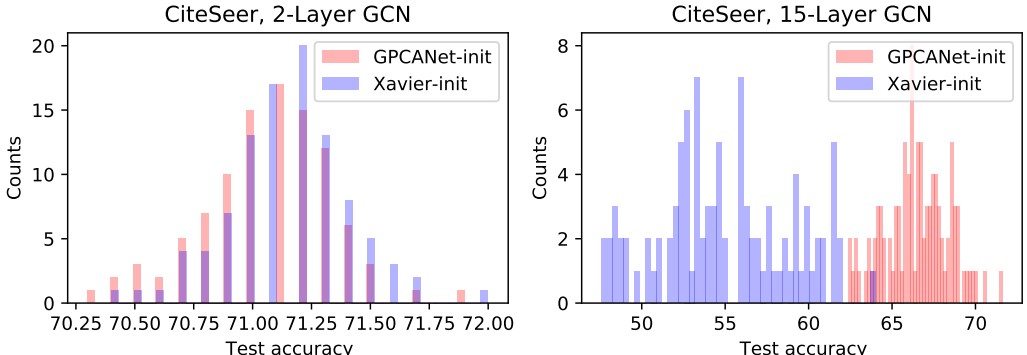

Figure 4: Comparison between Xavier-init and GPCANET-init in terms of test accuracy robustness over 100 seeds on CITESEER.

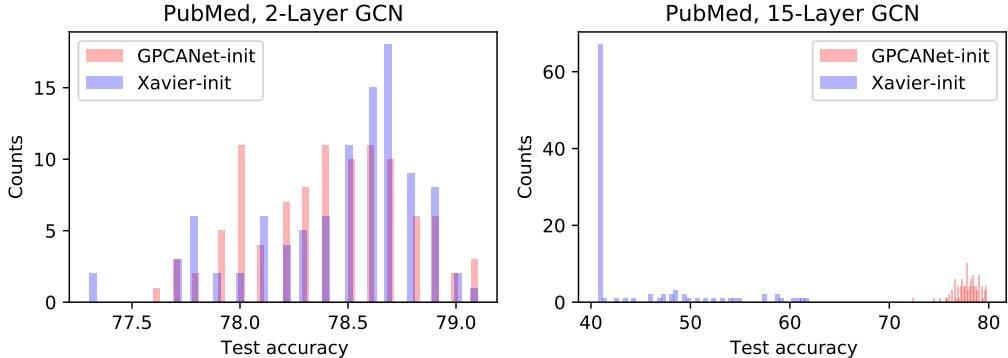

Figure 5: Comparison between Xavier-init and GPCANET-init in terms of test accuracy robustness over 100 seeds on PUBMED.

## A.9 TRAINING CURVE COMPARISON FOR GPCANET-INIT

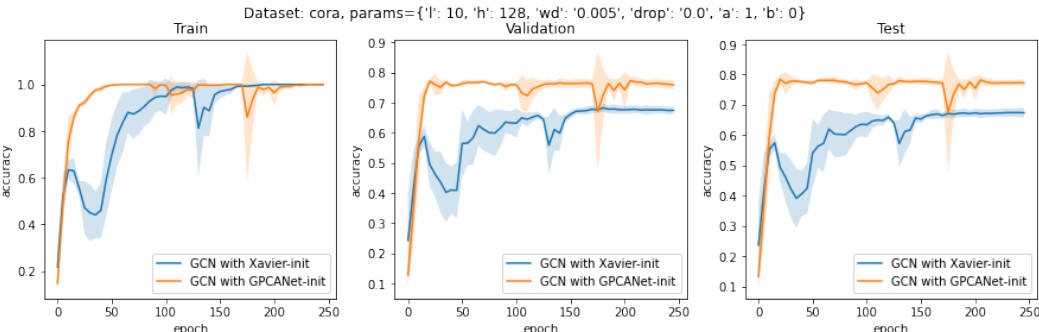

Figure 6: Training curve of 10-layer GCN initialized with Xavier initialization and GPCANET-Init on CORA.

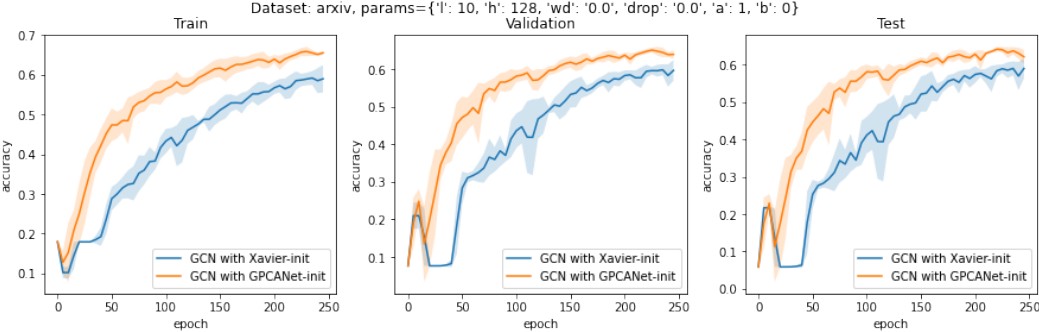

Figure 7: Training curve of 10-layer GCN initialized with Xavier initialization and GPCANET-Init on ARXIV.

## A.10 RUNTIME COMPARISON

We have analyzed the runtime complexity of GPCA, GPCANET, and GPCANET-Init in Sec.3.6 and show their runtime is linear in number of nodes. Table 8 presents the practical runtime comparison among all methods, measured in seconds/epoch for all models, and total initialization seconds for GPCANET-Init, which verified the complexity analysis. Besides, GPCA is a extremely fast method with strong performance, and should be used as a strong baseline in future research.

Table 8: Runtime comparison for different methods over all datasets.

|  | CORA | CITESEER | PUBMED | ARXIV | PRODUCTS |
|---|---|---|---|---|---|
| Num Nodes $n$ | 2,708 | 3,327 | 19,717 | 169,343 | 2,449,029 |
| Features $d$ | 1,433 | 3,708 | 500 | 128 | 100 |
| GCN (seconds/epoch) | 0.0025 | 0.0025 | 0.0040 | 0.0469 | 30.9544 |
| GPCA (seconds/epoch) | 0.0010 | 0.0010 | 0.0010 | 0.0072 | 0.0443 |
| GPCANET (seconds/epoch) | 0.0062 | 0.0101 | 0.0202 | 0.2172 | 31.3664 |
| GPCANET-Init (seconds) | 0.836 | 1.614 | 0.659 | 0.657 | 2.477 |

