# OpenReview forum: "Connecting Graph Convolution and Graph PCA"
_ICLR.cc/2022/Conference — ICLR 2022 Submitted_

### Official Review · Reviewer_eCjq · 2021-10-26

**Correctness:** 4
**Technical Novelty And Significance:** 2
**Empirical Novelty And Significance:** 2
**Recommendation:** 5
**Confidence:** 4

**Details Of Ethics Concerns:**

Nil

**Main Review:**

The contributions presented in this manuscript are quite obvious to my knowledge, but not very novel. Basically the paper is well-presented and written with great readability.  I cannot see any obvious flaws in the theory, analysis and experiments presented in this paper. In addition, the paper provides sufficient experimental evidence to back the main claims introduced by the authors.
The paper builds around the careful revisit to the classic graph-regularized PCA (GPCA). This observation is very interesting indeed. GPCA combines the PCA and the graph-structure information when building embedding (or latent representation) of the data, i.e., the feature linear transformation is restricted to the orthogonal projection. In fact, the PPNP is a similar way without restricting to the orthogonal feature transformation. In theory, for both GCN and PPNP, it is naturally to consider orthogonal transformation (parameter W) to make them as a constrained network, thus GPCANet becomes this special case. From this aspect, the novelty of the paper is limited. Similarly the idea of the semi-supervised version is not absolutely new, this has been long in application in supervised dimensionality reduction framework.

**Summary Of The Paper:**

This manuscript looks at the classic graph-regularized PCA (GPCA) and try to build the connection with GPCA and the state-of-the-art GCN, and finally proposes a new deep graph network GPCANet.  The authors make a number of clear contributions as listed in the paper: 1) they build the connection between GPCA and GCN, 2) Based on this connection, they propose novel way of using such GPCA as a graph layer or as an initialization process for training GCN etc. and 3) thus present a new architecture of GPCANet that performs well on their tests.

**Summary Of The Review:**

In general, there is no theoretical flaw in the paper, and the paper is well presented. However the novelty is limited as the proposed model is a parameter constrained NNPN or GCN.

---

> ### Author Response · Authors · 2021-11-19
> **Response to Reviewer eCjq**
>
> We thank the reviewer for the feedback, we would like to address the reviewer’s comments one by one.
>
> >In theory, for both GCN and PPNP, it is natural to consider orthogonal transformation (parameter W) to make them as a constrained network, thus GPCANet becomes this special case.
>
> Our paper is not about presenting a new method designed from constraining GCN’s weight with orthogonal constraint and claiming it is better: we never constraint the weight of GPCANet to be orthogonal, and the design of GPCANet is not the main contribution of the paper. The paper’s main contribution is **establishing the connection between GPCA and GCN theoretically, empirically verifying the connection, and exploring the implications of the connection in different ways: the design of GPCANet and the promising GPCANet-initialization for GNNs.**
>
> >Similarly the idea of the semi-supervised version is not absolutely new, this has been long in application in supervised dimensionality reduction framework.
>
> GPCA originally only has an unsupervised definition. We are the first to propose a supervised formulation. The design of supervised-GPCA is originally motivated from partial least squares (PLS, [1]) and linear discriminant analysis (LDA, [2]) which are some well-known supervised dimensionality reduction techniques. They do share similar insight of using correlation between features and labels, however applying the insight to GPCA to derive supervised-GPCA is still novel. Moreover, we also mathematically show that this corresponds to the addition of edges between same-label nodes (which is often done heuristically, without justification), which is not presented before.
>
> >The contributions presented in this manuscript are quite obvious to my knowledge, but not very novel.
>
> We understand that the reviewer may find many concepts mentioned in our paper familiar -- e.g. graph smoothing regularization, orthogonality constraints, supervised dimensionality reduction. We do however, would like to underline fully-novel aspects of our paper:
>
> 1. The mathematical connection between GCN and GPCA was not known before
> 2. We adapt the insight from LDA & PLS to derive supervised-GPCA that is not presented before.
> 3. We capitalize on connection 1. to: Stacking GPCA model, GPCANet
> 4. We also capitalize on GPCANet to propose the **first initialization method for GNNs**. The preliminary result of applying GPCANet-Init to GCN guides a promising direction of designing initialization methods for GNNs.
>
> Arguably, these are all non-trivial novel contributions of our paper which we show are not limited to theory, but have practical value.

---

> > ### Comment · Reviewer_eCjq · 2021-11-24
> > **Limited Novelty**
> >
> > Thank you Authors very much for your responses on my comments.  I echo YcNy for the comment too.   My view is we are not criticizing the novelty or contribution, but its limitation, not significant, just an incremental. Yes, the mathematical connection between GCN and GPCA is not mentioned in literature, but it is trivial by adding orthogonal constraint, just like we can add the so-called Stiefel manifold constraint to linear transformation weights in each layer of deep structure.

---

> > > ### Author Response · Authors · 2021-11-29
> > > **Response to orthogonal constraint argument**
> > >
> > > We thank the reviewer for the additional feedback. We would like to ask the reviewer one question: if the connection between GPCA and GCN is just by adding the orthogonal constraint to graph convolution operation, how can you derive, analytically in closed form, the weight matrix of GCN only based on the orthogonal constraint? We are a bit confused with the reviewer's argument about orthogonal constraint. The weight matrix calculated based on Theorem 3.2 is only possible to be derived based on GPCA, and GPCA is not orthogonal constraint + graph convolution. In fact, if we split the GPCA to several components, then it contains reconstruction, Laplacian regularization, and orthogonal constraint. We thank the reviewer, but we believe the connection is not trivial as the reviewer stated by "just adding orthogonal constraint".

---

### Official Review · Reviewer_YcNy · 2021-11-01

**Correctness:** 3
**Technical Novelty And Significance:** 2
**Empirical Novelty And Significance:** 2
**Recommendation:** 5
**Confidence:** 5

**Main Review:**

### Reasons to accept
1. The paper shows us that regularization of weights may be necessary for GNNs.
2. A new mixhop network architecture has been proposed that seems to be more promising than mixup performance.
3. This paper is well-written and easy to follow.

### Reason to reject
In general, I think the novelty of this article is insufficient mainly in the following points.
1. GPCA is only GC Operation with an orthogonal constraint. Eq (5) ||X-ZW^\top||_F^2+\alpha\Tr{Z^\top L Z} is able to reformulated as:
||XW-ZW^\topW||_F^2+\alpha\Tr{Z^\top L Z}. Because of W^\topW=I, we can have a GC operator defined in (1). So all things in this paper are based on the GC operator with the orthogonal constraint.
2. The definition of supervised formulation can hardly be considered as an original definition, and such methods are widely used in semi-supervised manifold learning (so much relevant literature), but this paper does not seem to be a suitable reference for the relevant methods.
3. The performance of the algorithm, as I analyzed earlier, is not very different and has significant advantages.

### Recommendation
1. Unless authors can theoretically analyze what benefits the orthogonal constraint actually brings, it is really hard to evaluate the contribution of this paper.
2. cite papers about semi-supervised (or supervised) manifold learning.

(1) Zhu and et, al Interpreting and Unifying Graph Neural Networks with An Optimization Framework, WWW 2021.


**Summary Of The Paper:**

This paper attempts to establish the relationship between Graph PCA and Graph Convolutional Layer, so as to define a new Graph Neural Network based on graph PCA.

**Summary Of The Review:**

Basically, the whole paper is based on posing an orthogonal constraint on GCN. However, they do not provide any convincing theoretical justification for that. Although it is a good paper to read, I tend to reject this paper because of a lack of novelty.

---

> ### Author Response · Authors · 2021-11-19
> **Response to Reviewer YcNy 1/2**
>
> We thank the reviewer for detailed reviews, we answer the reviewer’s questions one by one.
>
> First we would like to reply to the reviewer’s summary:
>
> >Basically, the whole paper is based on posing an orthogonal constraint on GCN. However, they do not provide any convincing theoretical justification for that.
>
> Our paper has 4 contributions: 1) mathematical connection between GPCA and GCN, 2) making GPCA supervised, 3) design of GPCANet, and 4) initialization of GCN through GPCANet. We respectfully disagree that the reviewer argues “the whole paper is only based on an orthogonal constraint on GCN”. Firstly, our paper is not about presenting a new method designed from constraining GCN’s weight with orthogonal constraint and claiming it is better: we never constraint the weight of GPCANet to be orthogonal, and the design of GPCANet is not the main contribution of the paper. The paper’s main contribution is **establishing the connection between GPCA and GCN theoretically, empirically verifying the connection, and exploring the implications of the connection in different ways: the design of GPCANet and the promising GPCANet-initialization for GNNs.**
>
> Also, we do provide theoretical justification for the connection between GPCA and GCN in Section 3. We also provide an insight of why (unsupervised) GPCA provides good weight to (supervised) GCN based on the relationship between (unsupervised) PCA and (supervised) 1-layer MLP. When working on dimensionality reduction, it’s easy to show that the 1-layer MLP trained with reconstruction loss (autoencoder with both encoder and decoder as 1-layer MLP) is very similar to PCA,  where the only difference is the orthogonality assumption. Then stacking PCA has a connection with MLP with reconstruction loss. Thus intuitively the PCA-calculated weight matrix at each layer is close to the weight matrix from MLP trained with reconstruction loss, which serves as an unsupervised pre-training task for supervised downstreaming tasks. Another way to think is that PCA preserves most information (largest variance = highest entropy in information theory) from input to its output, hence with PCA-calculated weight matrix applied to MLP at each layer, MLP preserves most information from 1st layer to the last layer.
> Coming back to GPCA and GCN, the solution of GPCA shares similarity with 1-layer GCN trained with reconstruction loss (graph autoencoder with encoder as 1-layer GCN and decoder as 1-layer MLP), hence the GPCA-calculated weight matrix may be close to the weight matrix from GCN trained with reconstruction loss. From another perspective, GPCA preserves most information from input to its output with additional graph smoothing requirements over output, which is useful for node-classification tasks. Hence with the GPCA-calculated weight matrix used as initialization of GCN at each layer, GCN preserves most information from 1st layer to the last layer and also takes the smoothness requirement over output into consideration.
>
>
> >GPCA is only GC Operation with an orthogonal constraint. Eq (5) $||X-ZW^T||_F^2+\alpha Tr(Z^T L Z)$ is able to reformulated as: $||XW-ZW^TW||_F^2+\alpha Tr(Z^T L Z)$. Because of $W^TW=I$, we can have a GC operator defined in (1). So all things in this paper are based on the GC operator with the orthogonal constraint.
>
> We believe the reviewer’s claim that two equations are the same is incorrect: right multiplication with $W$ cannot derive the formulation and the optimization objective shouldn’t multiply $W$ as it has trivial solution with $W=0$. Also we have to state that GCN does not have orthogonality constraints on columns of W, and we do not propose to enforce it either.
>
> >The definition of supervised formulation can hardly be considered as an original definition, and such methods are widely used in semi-supervised manifold learning.
>
> We respectfully disagree. GPCA originally only has an unsupervised definition. We are the first to propose a supervised formulation. The design of supervised-GPCA is originally motivated from partial least squares (PLS, [1]) and linear discriminant analysis (LDA, [2]) which are some well-known supervised dimensionality reduction techniques. They do share similar insight of using correlation between features and labels, however applying the insight to GPCA to derive supervised-GPCA is still novel. Moreover, we also mathematically show that this corresponds to the addition of edges between same-label nodes (which is often done heuristically, without justification), which is not presented before. We would be happy to closely study specific references the reviewer could provide to clarify the differences. We also have revised the manuscript to cite PLS and LDA.

---

> > ### Author Response · Authors · 2021-11-19
> > **Response to Reviewer YcNy 2/2**
> >
> > >The performance of the algorithm, as I analyzed earlier, is not very different and has significant advantages.
> >
> > First, we remark that GPCANet is only 1 of 4 contributions we have, where GPCANet’s performance is still significantly better than all baselines in large datasets (Arxiv and Products) from Open Graph Benchmark. Second, the great performance of GPCA and its supervised version deserve a highlight: GPCA is unsupervised and super fast (700x faster than GCN on Products, see the runtime table below.), and should serve as a strong baseline in future. Last but not least, the GPCANet-initialization shows significant improvement when training deeper models, which is also the **first proposed initialization method for GNNs**. We believe this work can guide future research in investigating specific initialization techniques of GNNs.
> >
> > We provide additional training and test wallclock time comparison among GCN, GPCA, GPCANet, and GPCANet-Initialization, where we use 5 layers, hidden size 128 for all models. Training time is measured in seconds/epoch, initialization is measured as total time in seconds.
> >
> > |Methods  |Cora|	Citeseer|	PubMed|	OGBN-Arxiv|	OGBN-Products|
> > |----|----|----|----|----|----|
> > |Num Nodes $n$| 	2,708|	3,327|	19,717	|169,343|	2,449,029|
> > |Features $d$|	1,433|	3,708|	500|	128|	100|
> > ||
> > |GCN (seconds/epoch) |	0.0025|	0.0025|	0.0040|	0.0469|	30.9544|
> > |GPCA (seconds/epoch) |	0.0010|	0.0010|	0.0010|	0.0072|	0.0443|
> > |GPCANet (seconds/epoch) |	0.0062|	0.0101|	0.0202|	0.2172|	31.3664|
> > ||
> > |GPCANET-Init (seconds) | 0.836 | 1.614 |0.659 |0.657 |2.477|
> >
> >
> > **Reference**:
> >
> > [1] Geladi, Paul, and Bruce R. Kowalski. "Partial least-squares regression: a tutorial." Analytica chimica acta 185 (1986): 1-17.
> >
> > [2] Balakrishnama, Suresh, and Aravind Ganapathiraju. "Linear discriminant analysis-a brief tutorial." Institute for Signal and Information Processing 18.1998 (1998): 1-8.
> >
> > ---------------
> > We thank the reviewer for the informative feedback and questions, we kindly ask the reviewer to consider raising the score if we properly addressed the reviewer’s concern.

---

> > ### Comment · Reviewer_YcNy · 2021-11-24
> > **Response to authors**
> >
> > For the first response: Although the author emphasizes four different contributions. we know that the next three contributions are derivations of the first. If the first one is not novel, then the next three are trivial.
> >
> > For the second response: I have said $W^TW=I$, why it has a trivial solution？ I just want to mention the proposed method is a GC operator with an orthogonal constraint.
> >
> > For the third response, LDA is also represented in the manifold learning framework. Please revise manifold learning papers. Compared with LDA, manifold learning is much closer than graph convolutional networks.

---

> > > ### Author Response · Authors · 2021-11-29
> > > **Response to novelty of the established connection**
> > >
> > > The reviewer consistently argues the connection between GPCA and graph convolution is trivial, because "the proposed method is a GC operator with an orthogonal constraint". We would like to ask the reviewer one question: if the connection between GPCA and GCN is just by adding the orthogonal constraint to graph convolution operation, how can you derive, analytically in closed form, the weight matrix of GCN only based on the orthogonal constraint? We are a bit confused with the reviewer's argument about orthogonal constraint. The weight matrix calculated based on Theorem 3.2 is only possible to be derived based on GPCA, and GPCA is not orthogonal constraint + graph convolution. In fact, if we split the GPCA to several components, then it contains reconstruction, Laplacian regularization, and orthogonal constraint.

---

### Official Review · Reviewer_Awat · 2021-11-02

**Correctness:** 4
**Technical Novelty And Significance:** 3
**Empirical Novelty And Significance:** 2
**Recommendation:** 6
**Confidence:** 3

**Main Review:**

The proposed method is technically sound.  The relation between PCA and GCN is well established in this paper.  My concerns are as follows,

1. The experimental results of GPCANET-INIT is insignificant. GPCANET-INIT performs very close to XAIVER-INIT when L=2 and L=3. Although GPCANET-INIT performs better than  XAIVER-INIT as the number of layers increase, it is not meaningful to compare them since the performance does not get better when the model gets deeper.
2. The paper lacks a unified ablation study. It is unclear to me which part of the proposed model contribute most to the model performance.
3. The computation cost of the proposed method is expensive due to eigenvalue decomposition.


**Summary Of The Paper:**

This paper relates GCN to PCA from the perspective of optimization. The authors propose Graph PCA that is a general form of GCN. They further introduce a regularization term that enforces nodes with same labels close to each other.

**Summary Of The Review:**

The biggest contribution of this paper is that it derives a more general GCN from the perspective of PCA.  The paper is well written and easy to follow. However, due to computation bottleneck and insignificant empirical results, I would rate this paper with a weak accept.

---

> ### Author Response · Authors · 2021-11-19
> **Response to Reviewer Awat**
>
> We thank the reviewer for the feedback. We address the reviewer’s questions one by one as follows.
>
> >The experimental results of GPCANET-INIT are insignificant. GPCANET-INIT performs very close to XAIVER-INIT when L=2 and L=3. Although GPCANET-INIT performs better than XAIVER-INIT as the number of layers increases, it is not meaningful to compare them since the performance does not get better when the model gets deeper.
>
> As deep networks are harder to train, good initialization is important for them, that’s why Xaiver initialization is proposed for **deep** networks and widely used in practice. We agree with the reviewer that the real-world benchmark datasets we used do not benefit from depth greater than 2-3, and for shadow GNNs, initialization does not improve much because shadow networks are easy to train (this is also true for other non-graph neural networks). Nevertheless, deep GNNs is an important direction with great potential (see the benefit of training 1000-layer GNN in [1]), and the result from Table 3 and Figure 2 speaks to the great strengths and potential of our proposed initialization **over Xaiver initialization** for deep GNNs. While current datasets do not seem to benefit from depth, GPCANET-INIT remains to provide great potential for other datasets thay may in the future, and serves as a great tool for researchers to investigate training deep graph neural networks and remind researchers that initialization matters.
>
> >The paper lacks a unified ablation study. It is unclear to me which part of the proposed model contributes most to the model performance.
>
> Our paper has 4 contributions: 1. mathematical connection between GPCA and GCN, 2. making GPCA supervised, 3. design of GPCANet, and 4. initialization of GCN through GPCANet.
> Since we are not sure by “proposed model” what the reviewer is referring to, we summarize ablation studies related to various parts:
>
> 1. For ablation study of GPCA, we extensively investigated the impact of $\alpha$ in Appendix.A.6 Table 6 for all 5 datasets we used, (where we find).
> 2. For ablation study of supervised GPCA, we show results with respect to different $\beta$ in Table 2.
> 3. For ablation study of GPCANet, we studied different combinations of $\alpha$ and the number of layers for GPCANet in Figure 1.
> 4. For initialization, we do not have anything to ablate since our theory suggests $\alpha=1$ and $\beta=0$.
>
> We will be happy to include any other specific additional ablation study the reviewer would like to suggest.
>
> >The computation cost of the proposed method is expensive due to eigenvalue decomposition.
>
> We have to reemphasize that the eigendecomposition is over **$d$ by $d$** matrix where $d$ is the feature or hidden size (like 128). As $d$ is a small and constant value, the eigendecomposition is super fast. What’s more, we compute $Z^*$ and $W^*$ *efficiently* with clever approximations. We refer the reviewer to Section 3.6, which describes the computational steps in detail -- the overall complexity is **linear** in graph size, i.e. the number of nodes and edges.
>
> More concretely, We provide additional training and test wallclock time comparison among GCN, GPCA, GPCANet, and GPCANet-Initialization, where we use 5 layers, hidden size 128 for all models. Training time is measured in seconds/epoch, initialization is measured as total time in seconds.
>
> |Methods  |Cora|	Citeseer|	PubMed|	OGBN-Arxiv|	OGBN-Products|
> |----|----|----|----|----|----|
> |Num Nodes $n$| 	2,708|	3,327|	19,717	|169,343|	2,449,029|
> |Features $d$|	1,433|	3,708|	500|	128|	100|
> ||
> |GCN (seconds/epoch) |	0.0025|	0.0025|	0.0040|	0.0469|	30.9544|
> |GPCA (seconds/epoch) |	0.0010|	0.0010|	0.0010|	0.0072|	0.0443|
> |GPCANet (seconds/epoch) |	0.0062|	0.0101|	0.0202|	0.2172|	31.3664|
> ||
> |GPCANET-Init (seconds) | 0.836 | 1.614 |0.659 |0.657 |2.477|
>
> We can see that GPCANet has similar running time to GCN, not slower. Moreover, GPCA is super fast (GPCA is around 700x times faster than GCN on OGBN-Products). Lastly, the total initialization time is always less than 3 seconds even for large datasets like OGBG-Products.
>
>
> **Reference:**
>
> [1] Li, Guohao, et al. 2021 "Training Graph Neural Networks with 1000 Layers."
>
> ------------
> We thank the reviewer for the informative feedback and questions, we kindly ask the reviewer to consider raising the score if we properly addressed the reviewer’s concern.

---

### Official Review · Reviewer_uURT · 2021-11-03

**Correctness:** 2
**Technical Novelty And Significance:** 2
**Empirical Novelty And Significance:** 4
**Recommendation:** 5
**Confidence:** 4

**Main Review:**

The biggest strength of this paper is the strong experimental results on their new initializations scheme on deep GNNs. I think that result alone could be the basis of a decent paper, if other major concerns are addressed.

Below is a major concern that the authors need to address:

1. the first order approximation statement (in the sense of the first order Taylor series approximation) only holds if the learned parameters W in the graph NN are the eigenvectors of X^T \phi_\alpha^{-1}X. If the learned parameters W in the NN are not the appropriate eigenvectors, then there is no meaningful first order approximation going on. The authors did not provide any justification for why the weights of the NN will be the eigenvectors of said matrix (though this is an interesting conjecture given that GPCA provides good initialization). I find the authors first-order approximation claim, at least as stated on a theoretical level, quite inaccurate. Perhaps a more accurate description be "there exist some parameter W for the GCN under which the GCN convolutional operator becomes a first order approximation of GPCA". This would be a much weaker (though more accurate) statement of the authors' claims.



**Summary Of The Paper:**

The authors establishes a connection between GPCA and graph convolution (graph convolution as a first order approximation of GPCA), and uses that connection to 1. propose a GPCANet model and 2. propose a new initialization strategy for GNNs.

**Summary Of The Review:**

Nice experimental results. Theoretical claim misleading/inaccurate as stated. If theoretical claim is fixed, then this could be a good paper.

---

> ### Author Response · Authors · 2021-11-19
> **Response to Reviewer uURT**
>
> We thank the reviewer for their positive feedback on the contributions of our paper. We address the reviewer’s concern in the following.
>
> >I find the author's first-order approximation claim, at least as stated on a theoretical level, quite inaccurate. Perhaps a more accurate description would be "there exist some parameter W for the GCN under which the GCN convolutional operator becomes a first order approximation of GPCA".
>
> Firstly, the reviewer’s rephrase that “there exists some parameter W for the GCN under which the GCN convolutional operator becomes a first order approximation of GPCA” is correct, which is implied by the Theorem 3.1 and first-order approximation. Second, We would like to emphasize a clarification: we do *not* claim that GPCA is an **approximation** of GCN. In the end of section 3.3, we stated: “the first-order approximation of (unsupervised) GPCA with alpha = 1 can be viewed as a graph convolution with a **fixed, data-driven** W”. This statement remains to be accurate. We also state: “graph convolution operation in GCN can be viewed as the first-order approximation of GPCA with alpha= 1 with a **learnable** W.” Both of the above statements do not conflict with the reviewer’s rephrase. We also added the reviewer’s suggestion in page 4, marked in blue.
>
> >The authors did not provide any justification for why the weights of the NN will be the eigenvectors of said matrix (though this is an interesting conjecture given that GPCA provides good initialization).
>
> We provide an insight of why (unsupervised) GPCA provides good weight to (supervised) GCN based on the relationship between (unsupervised) PCA and (supervised) 1-layer MLP. When working on dimensionality reduction, it’s easy to show that the 1-layer MLP trained with reconstruction loss (autoencoder with both encoder and decoder as 1-layer MLP) is very similar to PCA,  where the only difference is the orthogonality assumption. Then stacking PCA has a connection with MLP with reconstruction loss. Thus intuitively the PCA-calculated weight matrix at each layer is close to the weight matrix from MLP trained with reconstruction loss, which serves as an unsupervised pre-training task for supervised downstreaming tasks. Another way to think is that PCA preserves most information (largest variance = highest entropy in information theory) from input to its output, hence with PCA-calculated weight matrix applied to MLP at each layer, MLP preserves most information from 1st layer to the last layer.
> Coming back to GPCA and GCN, the solution of GPCA shares similarity with 1-layer GCN trained with reconstruction loss (graph autoencoder with encoder as 1-layer GCN and decoder as 1-layer MLP), hence the GPCA-calculated weight matrix may be close to the weight matrix from GCN trained with reconstruction loss. From another perspective, GPCA preserves most information from input to its output with additional graph smoothing requirements over output, which is useful for node-classification tasks. Hence with the GPCA-calculated weight matrix used as initialization of GCN at each layer, GCN preserves most information from 1st layer to the last layer and also takes the smoothness requirement over output into consideration.
> ---------------
> We thank the reviewer for the informative feedback and questions, we kindly ask the reviewer to consider raising the score if we properly addressed the reviewer’s concern.

---

### Decision · Program_Chairs · 2022-01-20

**Decision:**

Reject

**Comment:**

The paper presents several related results. The initial main result consists in relating GPCA to GCN, showing that GPCA can be understood as a first order approximation of some specific instance of GCN where the W matrix is directly defined on data. This result is then exploited to define a supervised version of GPCA. As a follow-up the authors propose a novel GPCA-based network (GPCANet) and a GPCANet initialisation for GNNs. The paper is well written and easy to read. Empirical results are reported to verify the above mentioned connection between  GPCA and GCN, as well as the performances of  GPCANet  and the proposed initialisation for GNNs. Overall, while the mentioned connection was never explicitly reported in the literature, its existence is not surprising and thus its significance seems to be limited. Also the performances of GPCANet do not seem to be significant from a statistical point of view. The novel initialisation procedure for GNNs seems to be interesting and promising, although the used datasets may not make evident its full power. Authors rebuttal and discussion did not change the reviewers' initial assessment.